# Neutralizing Antibodies against the SARS-CoV-2 Delta and Omicron BA.1 following Homologous CoronaVac Booster Vaccination

**DOI:** 10.3390/vaccines10122111

**Published:** 2022-12-09

**Authors:** Jianhua Li, Xiaoyan Li, Erqiang Wang, Jinye Yang, Jiaxuan Li, Chen Huang, Yanjun Zhang, Keda Chen

**Affiliations:** 1Zhejiang Provincial Center for Disease Control and Prevention, Hangzhou 310000, China; 2Sinovac Biotech Ltd., Beijing 100085, China; 3Shulan International Medical College, Zhejiang Shuren University, Hangzhou 310000, China

**Keywords:** neutralizing antibodies, CoronaVac, SARS-CoV-2, booster dose

## Abstract

Emerging severe acute respiratory syndrome coronavirus 2 (SARS-CoV-2) variants have reduced susceptibility to neutralization by vaccines. In response to the constantly updated variants, a global vaccine booster vaccination program has been launched. In this study, we detected neutralizing antibody levels against wild-type (WT), Delta (B1.617.2), and Omicron BA.1 viruses in serum after each dose of CoronaVac vaccination. We found that booster vaccination significantly increased the levels of neutralizing antibodies against WT, Delta, and Omicron BA.1. Compared with only one vaccination, neutralizing antibody levels increased by 19.2–21.6-fold after a booster vaccination, whilst two vaccinations only produced a 1.5–3.4-fold increase. Our results support the conclusion that the CoronaVac vaccine booster can increase neutralizing antibody levels and cross-reactivity and enhance the body’s ability to effectively resist the infection of new coronavirus variants, emphasizing the need for booster vaccination.

## 1. Introduction

Coronavirus disease 2019 (COVID-19) is a potentially severe acute respiratory infection caused by SARS-CoV-2, which has resulted in more than 580 million confirmed cases and more than 6 million confirmed deaths [1,2]. Two doses of COVID-19 vaccine were more than 70% effective against variants including Alpha (B.1.1.7) and Delta (B.1.617.2) before the Omicron (B.1.1.529) variant emerged globally in late 2021 [3,4,5]. Mutations (e.g., Q493R, N501Y, S371L, S373P, S375F, Q498R, and T478K) in the Omicron receptor-binding domain (RBD) region enhance virus binding to the angiotensin-converting enzyme 2 (ACE2), making it more infectious [6]. A study on the prevention of reinfection showed that the vaccine was 90.2% effective against Alpha, 85.7% against Beta, 92.0% against Delta, and only 56.0% against Omicron BA.1 [7], indicating that Omicron has a strong immune escape potential [8]. This also raised concerns about the efficacy of COVID-19 vaccines and neutralizing antibodies (NAbs) against Omicron.

Emerging SARS-CoV-2 variants continue to drive the global pandemic; therefore, the need for vaccines that provide efficacious and broad-spectrum protection has increased. Globally, over 30 vaccines based on different technologies and effects have been approved, 10 of which are World Health Organization (WHO)-approved. Most of these vaccines use the wild-type SARS-CoV-2 spike protein as the immunogen [9]. These vaccines have been very successful in evoking neutralizing humoral and cellular immunity, especially in decreasing COVID-19 infections, hospitalized cases, and deaths [10,11,12]. However, neutralizing antibody responses and vaccine efficacy vary with vaccine dose, decline over time after vaccination, and are adversely affected by new variants [13,14]. To counteract weakened antibody responses and new variant emergence, booster vaccine doses have been approved which are highly effective at inducing high NAb titers in vaccinated individuals [13]. To date, over 1.8 billion additional/booster immunizations have been administered globally [15].

CoronaVac is a 2-dose β-propiolactone-inactivated COVID-19 vaccine approved by WHO [16]. Many countries use two doses of CoronaVac as a priming immunization and a large-scale vaccination program giving the third dose of homologous inactivation to contain breakthrough infections of SARS-CoV-2 variants [17]. Therefore, it is important to evaluate the immunization effect of the CoronaVac booster to improve and update immunization strategies. However, considering the persistent mutations in COVID-19, sequential vaccination with different vaccine types can improve the breadth, strength, durability, and functionality of immune responses compared to boosting with the same type of vaccine. Increasing vaccination coverage for the third dose, initiating a fourth vaccination, and sequential immunization represent potential solutions to curb the COVID-19 pandemic.

## 2. Materials and Methods

### 2.1. Cell Culture

Vero cells (ATCC CCL-81, Sinovac Biotech, Beijing, China) were cultured in minimum essential medium (Gibco, Grand Island, NY, USA). Vero E6 (ATCC CRL-1586) and BHK-21-hACE2 cells stably expressing human ACE2, supplied by Prof. Xiao-Feng Qin, were cultured in high-glucose Dulbecco’s modified Eagle’s medium (Gibco). All media were supplemented with 10% fetal bovine serum (FBS, Gibco, USA), 1% penicillin-streptomycin, and 25 mM HEPES. The Vero cell medium was supplemented with 2 mM L-glutamine and all cells were passaged every 2–3 days using trypsin-EDTA (0.25%, Gibco).

### 2.2. Virus Stocks

Experiments were performed using three SARS-CoV-2 strains isolated at our Biosafety Level 3 virology laboratory (Zhejiang Provincial Center of Disease Control and Prevention, Hangzhou, China) [18]. SARS-CoV-2/Vero/WGF/2020/WZ122 (WT strain/EPI_ISL_12040150) and SARS-CoV-2/Vero/LXG/2021/ZJ28 (Delta/B.1.617.2/EPI_ISL_1911196) were isolated from a throat swab and cultured in Vero cells. SARS-CoV-2/VeroE6/DSh/2021ZJ25 (Omicron/B.1.1/EPI_ISL_12040149) was grown in Vero E6 cells. WT passage 3, Delta passage 5, and Omicron passage 3 virus-containing supernatants were harvested at 80% cytopathogenic efficiency (CPE), and viral titers were determined using a microdose CPE assay. Virus stocks were sequenced using Illumina NextSeq (Illumina Inc., San Diego, CS, USA) to verify that they contained the expected spike protein sequence, with no changes to the furin cleavage sites.

### 2.3. Blood Samples

Zhejiang Provincial Center for Disease Control and Prevention recruited 43 volunteers who received a three-dose CoronaVac homologous vaccination regimen between 29 March 2020 and 27 July 2021. The place of inoculation was Changhe Street Community Health Service Center, Temporary Vaccination Site for Fangcang in Binjiang District, Hangzhou. The 43 participants provided 129 blood samples at 28 days after each dose of the vaccine. Participants’ clinical information (age, gender, and physical fitness) was recorded at sampling. Serum samples obtained after centrifugation at 2000 rpm for 15 min were stored at −80 °C and inactivated at 56 °C for 30 min before use. 

### 2.4. Ethical Approval

The study protocol was approved by the Ethics Committee of Zhejiang Provincial Center of Disease Control and Prevention. 

### 2.5. Live Virus Neutralization Test

The neutralizing antibody titer test was carried out in July 2022. Post-vaccination serum samples were serially diluted 2-fold with cell culture medium and mixed 1:1 with 100 TCID_50_ (median tissue culture infectious dose)/50 µL virus suspension in a 96-well plate. After 2 h of incubation, 1–2 × 10^4^ Vero-E6 cells were added to the serum–virus mixture and incubated for 3 days at 37 ℃ in a 5% CO_2_ incubator. The CPE in each well was recorded under a microscope, and the neutralization titer was calculated by the dilution number of 50% of the protective condition.

### 2.6. Statistical Analysis

Data are presented as the mean ± SD. Statistical analyses were conducted using GraphPad Prism 9.4.1 (GraphPad Inc., La Jolla, CA, USA). Differences between independent samples and those between two related samples were evaluated using unpaired and two-tailed *t*-tests, respectively. *p* < 0.05 was considered significant (* *p* < 0.05, ** *p* < 0.01, *** *p* < 0.005, **** *p* < 0.001).

## 3. Results

### 3.1. SARS-CoV-2 OmicronBA.1 Is a Novel and Highly Mutated Variant

The S protein of SARS-CoV-2 consists of S1 and S2 subunits and is the major viral surface protein. The S protein can bind to human ACE2 and enters the host cell mediated by the C-terminal RBD of the S1 subunit [19,20,21]. Subsequently, the S2 subunit mediates the fusion of the viral envelope with the host cell membrane, resulting in the release of the viral genome into the cytoplasm [22,23]. Compared with the Delta variant and WT, Omicron BA.1 mutation sites are more abundant and widely distributed on the surface of the NTD and RBD domains, spanning the ACE2 binding site and NAb epitopes. These mutations offer a potential evolutionary advantage by enhancing viral RBD-ACE2-binding or immune escape from NAbs [24,25]. The RBD, which interacts with the ACE2 receptor, contains 15 of these mutations: G339D, S371L, S373P, S375F, K417N, N440K, G446S, S477N, T478K, E484A, Q493R, G496S, Q498R, N501Y, and Y505H (Figure 1) [26]. 

### 3.2. CoronaVac Vaccine Booster Is Effective in Increasing Neutralizing Antibody Levels and Cross-Reactivity

To determine the susceptibility of WT, Delta, and Omicron viruses to CoronaVac-induced neutralization, we examined antibody levels and neutralizing activity in serum after each vaccination with CoronaVac. Regardless of whether it was against the original strain or the SARS-CoV-2 variant, the third dose of inoculation significantly increased serum NAb titers. For the WT, NAb levels increased by 3.3-fold after two doses compared with one dose (geometric mean titer, GMT 26), and by 19.4-fold after three doses (Figure 2A). There were 21.6-fold and 19.2-fold increases in NAb titers against Delta (GMT 110) (Figure 2B) and Omicron BA.1 (GMT 45) (Figure 2C) after the third dose, respectively. At the same time, we also calculated the serum positivity for NAbs. CoronaVac is an inactivated vaccine designed for the WT; therefore, the different vaccine doses did not affect the proportion of serum antibodies in the WT, which were all 100% (Figure 2A). However, there were obvious changes in the live virus neutralization experiments using the SARS-CoV-2 variant strains Delta and Omicron BA.1. For Delta serum, positivity increased from 53.49% at one injection to 95.35% after two injections, reaching 100% after three injections (Figure 2B). For Omicron BA.1, serum positivity increased from 16.28% after the first dose to 93.02% after the third dose (Figure 2C). Therefore, we believe that the third homologous booster shot of the inactivated vaccine is immunogenic and can effectively improve the body’s neutralizing immune response to a SARS-CoV-2 variant. 

### 3.3. Neutralizing Antibody Responses to SARS-CoV-2 Variants Indicate Massive Escape of Omicron

Serum NAb results elicited by CoronaVac showed that the GMT for Omicron was 10.7–28.7-fold lower than that of the WT, and was significantly lower than that for the Delta variant (Figure 3A–C). Numerous site mutations in the RBD region confer the ability of Omicron to evade immunity from vaccines or previous infection and, more broadly than any other variant, impair the potency of NAbs, which also contributes to a significant decrease in the protective efficacy of existing vaccines against Omicron infection.

## 4. Discussion

This study collected blood samples from volunteers who received a three-dose CoronaVac homologous vaccination regimen and measured live virus cross-NAb levels after each dose. The results showed that the CoronaVac booster could significantly increase the level of NAbs against Delta and Omicron BA.1, enhance the serum positivity for antibodies, and increase the body’s ability to effectively defend against new coronavirus variants. This was consistent with previous research results showing that vaccination with homologous or heterologous boosters can increase neutralizing titers by severalfold [24,27,28]. Studies have shown that nearly all individuals with anti-SARS-CoV-2 CD8+ T cell responses recognize the Omicron variant, while established SARS-CoV-2 stimulatory CD4+ and CD8+ T cell responses to Omicron remain largely unchanged, suggesting that this variant does not produce widespread T-cell immune evasion [29,30]. This also implies that T-cell immunity, which is less susceptible to Omicron mutations, might still be the key to preventing infection. Previous reports have suggested that Omicron infections are mild in nature, with serious infections rarely occurring in fully vaccinated individuals [31,32]. Therefore, we believe that the third dose of inactivated vaccine should be administered to stimulate the development of humoral responses and improve the immune barrier, thereby enhancing neutralizing potency and breadth and reducing the severity and mortality in the population.

The T478K mutation, coexisting with other complex mutations, might be associated with direct enhancement of the interaction of the RBD region with the ACE2 receptor. Residue R493, located in the RBD region, substitutes the hydrogen bond with ACE2 residue E35 with a new salt bridge, while residue R498, in addition to keeping the hydrogen bond interaction between residue 498 and ACE2 residue Q42, also interacts with ACE2 residue D38 to form a new salt bridge [33]. RBD residue S496 added a new interaction by forming a hydrogen bond with ACE2 residue K353, while compensating for the loss of affinity for ACE2 caused by the K417N mutation [34,35]. N501Y will form an additional stacked ring with Y41 [36]. D614G existed in multiple mutant strains in the early stage, which is mainly related to membrane fusion and can promote fusion of the virus with the human cell membrane [37]. These mutations resulted in increased affinity of Omicron for hACE2 and a significant increase in infectivity, which was 10-fold and 2.8-fold higher than that of the WT and Delta variants, respectively [31].

Research from Oxford University demonstrated that the antigenic distance of Omicron BA.1 is farther than that of Delta from the original SARS-CoV-2 vaccine strain [38], and most current vaccines use the original wild-type SARS-CoV-2 spike protein as the sole immunogen which has drawn attention to specific variant vaccines. Several recent studies have generated and tested Omicron-specific vaccine candidates with different vaccine antigen designs and components. However, the ability of these Omicron-specific vaccines to trigger serum NAb production in the body shows that they have limited cross-protection ability against different variants of SARS-CoV-2, and do not have broad spectrum activity [39,40,41]. In addition, new variants of SARS-CoV-2 emerge every 6 months on average, making it unlikely that vaccines for new variants will be developed and completed promptly.

For SARS-CoV-2, the main goal of current vaccines is to prevent symptomatic COVID-19 and contain its rapid spread and worldwide epidemic. This study lacked evaluation of the cross-protection effect of the currently popular variants, such as Omicron BA.4/5. In the future, we will further study the protective efficiency of homologous vaccination with CoronaVac, and heterologous vaccination, such as CoronaVac combined with other types of vaccines (adenovirus vaccines, mRNA vaccines or recombinant protein vaccines) on subsequent circulating mutants, to find the best immunization regimen for current or potential future outbreaks.

## Figures and Tables

**Figure 1 vaccines-10-02111-f001:**
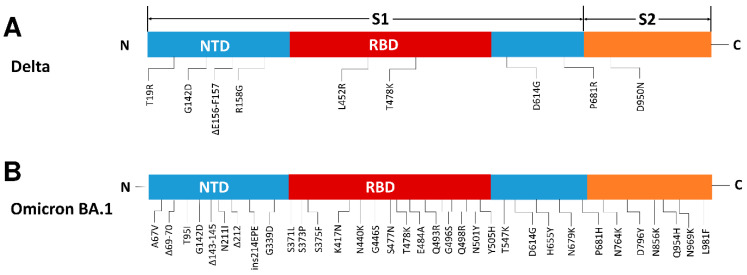
Schematic of severe acute respiratory syndrome coronavirus 2 (SARS-CoV-2) spike protein structure showing the mutations of the variants used in this study (https://outbreak.info (accessed on 10 August 2022)). The N-terminal domain (NTD) is shown in blue and red denotes the receptor-binding domain (RBD). (**A**) Linear mutation diagram of Delta spike (S1 and S2) proteins. (**B**) Linear mutation diagram of Omicron spike proteins. Mutations shared by Delta with Omicron are shown in red. Different mutations of Omicron and Delta at the same site are shown in blue.

**Figure 2 vaccines-10-02111-f002:**
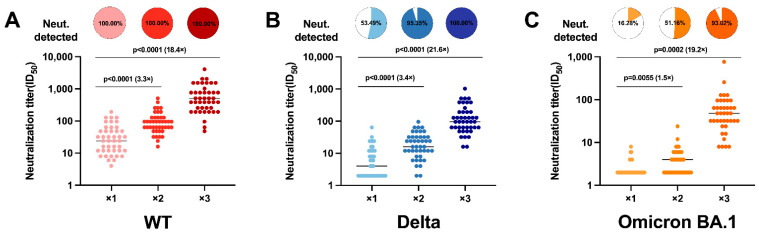
Live virus neutralization test results from 129 serum samples obtained from 43 volunteers vaccinated with three doses of CoronaVac homologously after each dose of CoronaVac. Neutralization of authentic viruses was performed using a cytopathic effect (CPE)-based assay with a viral titer of 10^10^ TCID_50_ (median tissue culture infectious dose). The neutralization titer of the serum sample was calculated as the reciprocal of the highest dilution that protected more than 50% of cells from CPE. Pie charts show the proportion of vaccinees within each group that had detectable neutralization against the severe acute respiratory syndrome coronavirus 2 (SARS-CoV-2) live virus. Fold-increase in the geometric mean neutralization titer of each dose relative to those vaccinated with one dose within a group is shown as a number with the “×” symbol above the lines. (A-C) The results of neutralizing antibodies against the wild-type (WT) (**A**), Delta (**B**), and Omicron BA.1 (**C**) viruses in the serum indicated by dots.

**Figure 3 vaccines-10-02111-f003:**
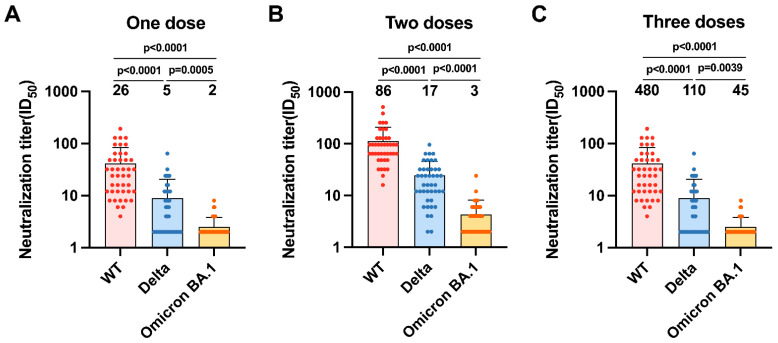
(**A**–**C**) The results of cross-neutralization of wild-type (WT), Delta, and Omicron BA.1 viruses by sera from each individual vaccinated with one (**A**), two (**B**), and three (**C**) doses of CoronaVac are shown as bar graphs. The numbers on the graph represent the geometric mean titers (GMT).

## Data Availability

The data presented in this study are available on request from the corresponding author.

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
