# Peer review of "Neutralizing Antibodies against the SARS-CoV-2 Delta and Omicron BA.1 following Homologous CoronaVac Booster Vaccination"

_vaccines, 2022, doi:10.3390/vaccines10122111_

Round 1

Reviewer 1 Report

This is an interesting and well-written study. 

I only have a few minor suggestions

1. Please clarify the timing of the study. At the start of the study (March 2020) the omicron variant did not exist. This is puzzling so please clarify.

2. Provide some (short) information on where this vaccine is in use

3. When mentioning >70% protection. Is this against infection? Ot (severe) disease?

4. "Curb virus spread". Really? Please include a reference.

5. Did any of the volunteers experience natural infection? How was this established?

Author Response

Point 1. Please clarify the timing of the study. At the start of the study (March 2020) the omicron variant did not exist. This is puzzling so please clarify.

Response 1: Thank you very much. We would like to thank you for your thoughtful and sincere review. We have added this sentence “The neutralizing antibody titer test was carried out in July 2022” in the neutralizing antibody test content of the methodology. The change has been highlighted in the revised manuscript.

Point 2. Provide some (short) information on where this vaccine is in use

Response 2: Thank you so much. The comments and suggestions made by you have helped us improve the manuscript significantly. We have revised it according to your recommendation and added it to lines 87-89.

Point 3. When mentioning >70% protection. Is this against infection? Ot (severe) disease?

Response 3: Thank you very much. We have revised it according to your suggestion.

Point 4. "Curb virus spread". Really? Please include a reference.

Response 4: Thank you very much. We have revised it according to you recommendation.

Point 5. Did any of the volunteers experience natural infection? How was this established?

Response 5: None of the volunteers participating in the study had been infected with SARS-CoV-2. The proportion of infected people in China is very low, accounting for only about 1%, so it is not difficult to find 43 volunteers who have not been infected with SARS-CoV-2 to participate in the study, and people who have been infected with SARS-CoV-2 in China have hospital admission records that can be easily traced.

Reviewer 2 Report

The communication entitled “Neutralizing antibodies against the SAARS-CoV-2 Delta and Omicron BA.1 following homologous Corona Vac booster vaccination” focuses on the efficacy of the vaccine booster vaccination in 43 individuals affected by the variant named Omicron BA.1 through the measurement of the antibody levels and neutralizing activity . The purpose of this study is corrected and the project study is well-conducted. However, this manuscript does not report original data with respect to previous publications about this topic. Therefore, I think that this article is not suitable for publication.

Author Response

Thank you very much. We collected human sera that had not been infected with SARS-CoV-2 and had completed homologous booster immunization with the CoronaVac and evaluated the levels of live virus-neutralizing antibodies against SARS-CoV-2 and its variants. Many countries have adopted CoronaVac's homologous booster immunization program to achieve the purpose of preventing breakthrough infection of SARS-CoV-2 variants. At present, there are relatively few studies on CoronaVac homologous booster immunization, most of which are pseudovirus-neutralizing antibody experiments, so our research has practical significance. At the same time, the prevalence of SARS-CoV-2 infection in China was extremely low, with only 1% of the population infected, thus providing clear information on volunteers with minimal confounding factors and high study reliability. We think our research can provide useful information on this topic.

Reviewer 3 Report

In this manuscript, Li et al., investigated the neutralizing titer after CoronaVac vaccination. They showed that CoronaVac vaccine booster increased the neutralizing antibody levels and cross-reactivity. Overall the experiments were well performed, but the authors should emphasize the novelty of this paper as there are similar paper about the neutralizing antibody post CoronaVac vaccination. I have some major and minor comments.

Major

1.     Figure 2, 3: Each ID50 (between x1, x2 x3 or between WT, Delta, BA.1) had significant difference? The authors should add statics analysis data in the Figure.

2.     Do the authors have any information if these donors had COVID-19 history before or after vaccination?

3.     Line 21-23: The authors concluded that “CoronaVac booster can increase neutralizing antibody level and cross-reactivity…”. Based on their data, although the booster increased neutralizing antibody, the titer against BA.1 was very low. Do the authors think booster is still effective to prevent BA.1 infection or just to prevent severe diseases? The authors should discuss and clarify this point.

4.     As the authors know, there are so many reports regarding neutralizing antibody titers after CoronaVAc (both homologous and heterologous). The authors should mention the novelty of this paper.

Minor

          1. Figure 2B: It’s hard to see the number in the dark blue circle. Please change the color.

Author Response

Major

Point 1. Figure 2, 3: Each ID50 (between x1, x2 x3 or between WT, Delta, BA.1) had significant difference? The authors should add statics analysis data in the Figure.

Response 1: Thank you very much. We have revised it.

Point 2. Do the authors have any information if these donors had COVID-19 history before or after vaccination?

Response 2: None of the volunteers participating in the study had been infected with SARS-CoV-2.

Point 3. Line 21-23: The authors concluded that “CoronaVac booster can increase neutralizing antibody level and cross-reactivity…”. Based on their data, although the booster increased neutralizing antibody, the titer against BA.1 was very low. Do the authors think the booster is still effective to prevent BA.1 infection or just to prevent severe diseases? The authors should discuss and clarify this point.

Response 3: Thank you very much. We have revised it according to you recommendation.

Point 4. As the authors know, there are so many reports regarding neutralizing antibody titers after CoronaVAc (both homologous and heterologous). The authors should mention the novelty of this paper.

Response 4: We collected human sera that had not been infected with SARS-CoV-2 and had completed homologous booster immunization with the CoronaVac and evaluated the levels of live virus-neutralizing antibodies against SARS-CoV-2 and its variants. At present, studies on homologous booster immunization with CoronaVac are relatively few and most of them are pseudo-viral neutralizing antibody experiments. At the same time, the prevalence of SARS-CoV-2 infection in China is extremely low, with only 1% of the population infected, thus providing clear information on volunteers, minimal confounding factors, and high study reliability.

Minor

Point 1. Figure 2B: It’s hard to see the number in the dark blue circle. Please change the color.

Response 1: Thank you very much. We have revised it.

Round 2

Reviewer 2 Report

In the revised version of the article entiitled “Neutralizing antibodies against the SARS-CoV-2 delta and Omicron BA.1 following homologous CoronaVac booster vaccination” the authors sufficiently replied to the comments of the reviewer, Anyway, I only suggest to add the article recently published in the N Engl J Med  (Atmar RL et al N Engl J Med 202; 386:1046-1057). I think that this revised version s suitable for publication in its current version.

Reviewer 3 Report

I think the authors answered all my questions and the they improved the manuscript.